# Demographic history mediates the effect of stratification on polygenic scores

**Arslan A Zaidi\*, Iain Mathieson\***

Department of Genetics, Perelman School of Medicine, University of Pennsylvania, Philadelphia, United States

**Abstract** Population stratification continues to bias the results of genome-wide association studies (GWAS). When these results are used to construct polygenic scores, even subtle biases can cumulatively lead to large errors. To study the effect of residual stratification, we simulated GWAS under realistic models of demographic history. We show that when population structure is recent, it cannot be corrected using principal components of common variants because they are uninformative about recent history. Consequently, polygenic scores are biased in that they recapitulate environmental structure. Principal components calculated from rare variants or identity-by-descent segments can correct this stratification for some types of environmental effects. While family-based studies are immune to stratification, the hybrid approach of ascertaining variants in GWAS but reestimating effect sizes in siblings reduces but does not eliminate stratification. We show that the effect of population stratification depends not only on allele frequencies and environmental structure but also on demographic history.

**\*For correspondence:**
aazaidi@pennmedicine.upenn.edu
(AAZ);
mathi@pennmedicine.upenn.edu
(IM)

**Competing interests:** The authors declare that no competing interests exist.

## Introduction

Population structure refers to patterns of genetic variation that arise due to non-random mating. If these patterns are correlated with environmental factors, they can lead to spurious associations and biased effect size estimates in genome-wide association studies (GWAS). Approaches such as genomic control (GC) (*Devlin and Roeder, 1999*), principal component analysis (PCA) (*Price et al., 2006*), linear mixed models (LMMs) (*Kang et al., 2010*; *Loh et al., 2015*) and linkage disequilibrium score regression (LDSC) (*Bulik-Sullivan et al., 2015a*) have been developed to detect and correct for this stratification. However, these approaches do not necessarily remove all stratification, particularly when multiple studies are meta-analyzed (*Berg et al., 2019*; *Sohail et al., 2019*). Large GWAS in relatively homogeneous populations, such as the UK Biobank (UKB) (*Bycroft et al., 2018*), should alleviate many of these concerns. However, such populations still exhibit fine-scale population structure (*Leslie et al., 2015*; *Karakachoff et al., 2015*; *Kerminen et al., 2017*; *Haworth et al., 2019*; *Raveane et al., 2019*; *Bycroft et al., 2019*; *Byrne et al., 2020*). The extent to which this fine structure impacts GWAS inference in practice is largely unknown, and it is not clear whether existing methods adequately correct for it. This question has become increasingly acute in light of the recent focus on polygenic scores for disease risk prediction (*Torkamani et al., 2018*; *Knowles and Ashley, 2018*). Polygenic scores for many physical and behavioral traits exhibit geographic clustering within the UK even after stringent correction for population structure (*Haworth et al., 2019*; *Abdellaoui et al., 2019*). Although some of this variation may be attributed to recent migration patterns (*Abdellaoui et al., 2019*), it could also reflect residual stratification in effect size estimates (*Lawson et al., 2020*).

To address these questions, we investigated the effect of population structure on GWAS in a simulated population with a similar degree of structure to the UK Biobank. We considered the fact that different demographic histories can give rise to the same overall degree of population structure (in terms of statistics such as $F_{ST}$ and the genomic inflation factor, $\lambda$). This is relevant because the

degree to which common and rare variants are impacted by, and are thus informative about, population structure depends on demographic history. It is therefore important to understand the demographic history of GWAS populations in order to assess the consequences of stratification.

## Results

### Rare variants capture recent population structure

We leveraged recent advances in our understanding of human history to simulate GWAS under different realistic demographic models. We simulated population structure using a six-by-six lattice-grid arrangement of demes with two different symmetric stepping-stone migration models (*Figure 1*). First, a model where the structure extends infinitely far back in time (perpetual structure model; e.g. *Mathieson and McVean, 2012*) and second, a model where the structure originated 100 generations ago (recent structure model). This second model is motivated by the observation from ancient DNA that Britain experienced an almost complete population replacement within the last 4,500 years (*Olalde et al., 2018*), providing an upper bound for the establishment of present-day geographic structure in Britain. We set the migration rates in the two models to match the degree of population structure in the UK Biobank, measured by the average $F_{ST}$ between regions (*Leslie et al., 2015*) and the genomic inflation factor for a GWAS of birthplace in individuals with 'White British' ancestry from the UK Biobank (*Haworth et al., 2019*).

Population structure in the two models is qualitatively different, even though $F_{ST}$ is the same. When structure is recent, it is driven largely by rare variants which tend to have a more recent origin (*Gravel et al., 2011*; *Fu et al., 2013*; *O'Connor et al., 2015*) and are therefore less likely to be shared among demes. Common variants, because they are older and usually predate the onset of structure in our model, are more likely to be shared among demes and have not drifted enough in 100 generations to capture the spatial structure effectively. Therefore, recent structure is captured by the principal components of rare variants (rare-PCA) but not common variants (common-PCA) (*Figure 1*). In fact, 100 common-PCs altogether explain only 3% of the variance in rare-PC1 (*Figure 1—figure supplement 1*). In comparison, when population structure is perpetual, both common and rare variants carry information about spatial structure (*Figure 1*, *Figure 1—figure supplements 1*, 100 common-PCs explain 50% of the variance in rare-PC1). The two models discussed here represent somewhat extreme demographic scenarios and in reality, the degree to which common and rare variants capture independent aspects of population structure will depend on how the structure varies through time (*Figure 1—figure supplement 1*).

PCA with rare variants requires sequence data. When only genotype data are available, imputed rare variants can be used *Figure 1—figure supplement 2*. However, the practical utility of this approach would depend on the imputation accuracy which in turn depends on the population, the imputation algorithm and the reference panel (*Das et al., 2018*). Another alternative is to carry out PCA on haplotype or identity-by-descent (IBD) sharing, which is also informative about recent population structure (*Figure 1—figure supplement 2*).

### The impact of population stratification depends on demographic history

That common variants fail to capture recent population structure has important implications for GWAS. Most GWAS use PCA or LMMs, both of which rely on the genetic relatedness matrix (GRM) to describe population structure. Since rare variants are not well-represented on SNP arrays, the GRM is usually constructed from common variants. This will lead to insufficient correction if common variants do not adequately capture recent population structure. To test this, we simulated a GWAS (N = 9,000) of a non-heritable phenotype (i.e. $h^2 = 0$) with an environmental component that is either smoothly (e.g. latitude) or sharply (e.g. local effects) distributed in space (*Figure 2*). We calculated GRMs using either common (minor allele frequency, MAF > 0.05) or rare variants (minor allele count, MAC = 2, 3, or 4), and included the first 100 PCs in the model to correct for population structure.

When population structure is recent, smooth environmental effects lead to an inflation in common, but not rare, variants and this inflation can only be corrected with rare- but not common-PCs (*Figure 2B*, top row). This is a consequence of the fact that rare variants carry more information about recent structure than common variants (*Figure 1*). We find similar results using LMMs instead

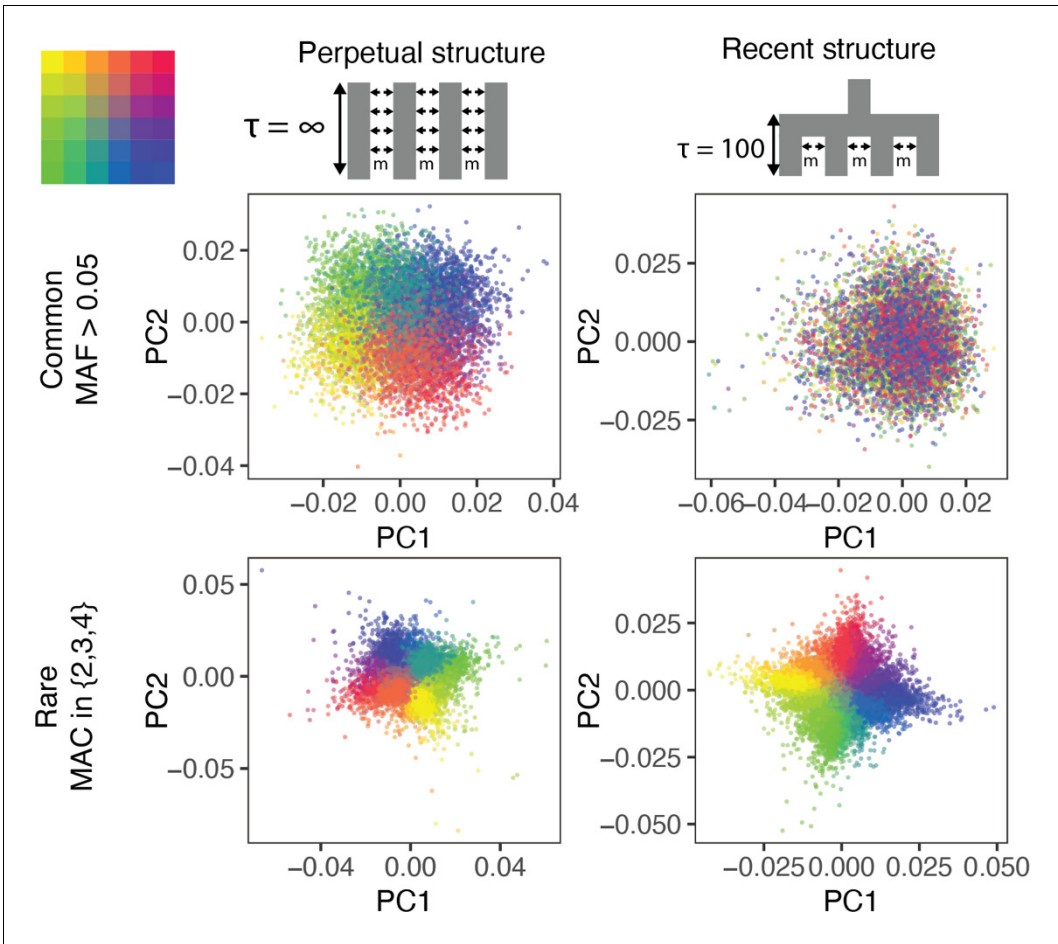

**Figure 1.** The ability of PCA to capture population structure depends on the frequency of the variants used and the demographic history of the sample. Panels show the first and second principal components (PCs) of the genetic relationship matrix constructed from either common (upper row) or rare (lower row) variants. Each point is an individual (N = 9,000) and their color represents the deme in the grid (upper left) from which they were sampled. Both common (minor allele frequency >0.05) and rare (minor allele count = 2, 3, or 4) variants can be informative when population structure is ancient (left column; $\tau = \infty$ represents the time in generations in the past at which structure disappears) but only rare variants are informative about recent population structure (right column; $\tau = 100$ generations). Number of variants used for PCA: 200,000 (upper row), 1 million (lower left), and $\approx 750,000$ (lower right).

The online version of this article includes the following figure supplement(s) for figure 1:

**Figure supplement 1.** Collinearity between common- and rare-PCs under different demographic models.

**Figure supplement 2.** PCA on imputed rare variants and IBD-sharing provide an alternative to rare-PCA when sequence data are not available.

---

of PCA (*Figure 2—figure supplement 1*). Therefore, in studies with recent structure, such as the UKB, neither PCA- nor LMM-based methods will fully correct for stratification as long as the GRM is derived from common variants. In contrast, under the perpetual structure model, both common and rare variants may be inflated due to smooth environmental effects (*Figure 2A*, top row), but this inflation is largely corrected with either common- or rare-PCs (*Figure 2A*, top row).

Local environmental effects largely impact rare variants only (*Mathieson and McVean, 2012*; *Figure 2A*, lower row) and the inflation due to local effects cannot be fully corrected using either common- or rare-PCs (*Figure 2A and B*, lower row). This is because local environmental effects cannot be represented by a linear combination of the first hundred principal components. Importantly, local effects only impact a small subset of variants—those clustered in the affected deme(s)—resulting in inflation only in the tails of the test statistic distribution (*Figure 2*). This pattern of inflation

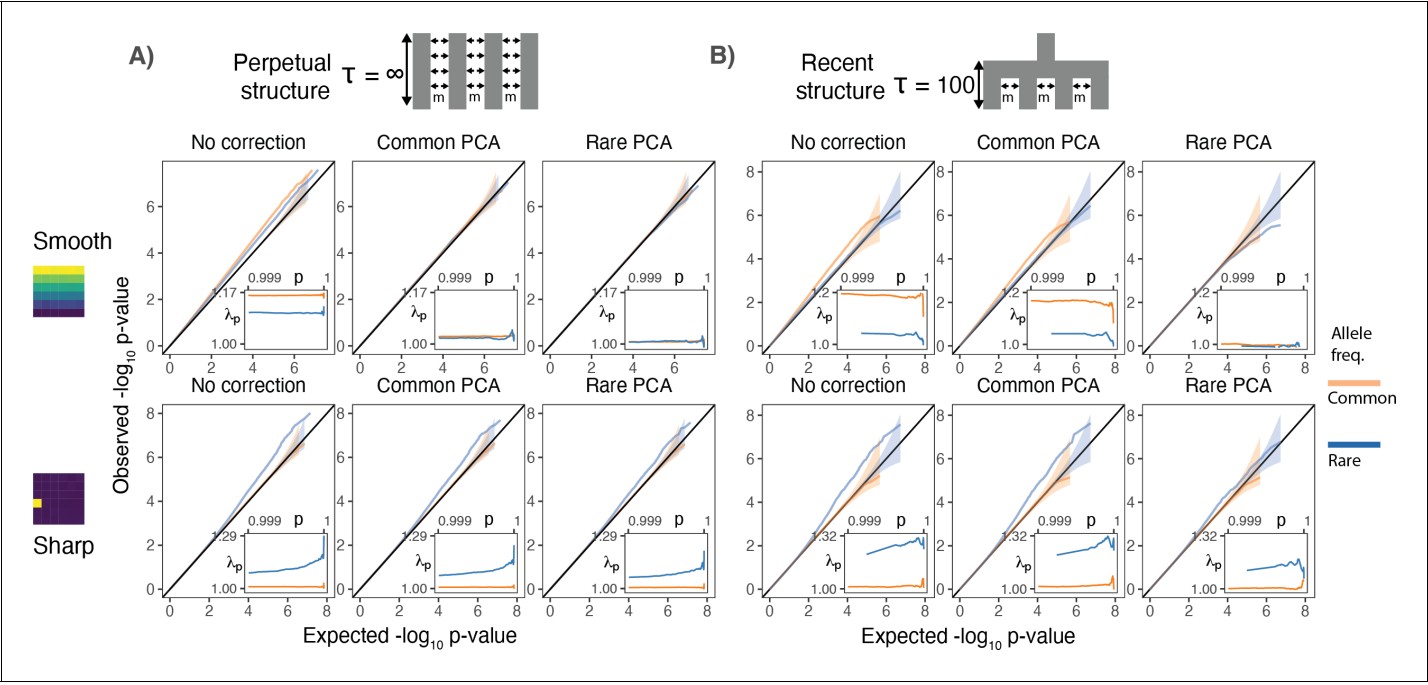

**Figure 2.** Test statistic inflation under two different demographic histories. (A) Perpetual structure and (B) recent structure. Upper and lower rows show results for smoothly and sharply distributed environmental risk, respectively, whereas columns show different methods of correction. The simulated phenotype has no genetic contribution so any deviation from the diagonal represents inflation in the test statistic. Each panel shows QQ plots for -log$_{10}$ p-value for common (orange) and rare (blue) variants. Insets show inflation ($\lambda_p$) in the tail (99.9th percentile) of the distribution. Results are averaged across 20 simulations of the phenotype.

The online version of this article includes the following figure supplement(s) for figure 2:

**Figure supplement 1.** QQplots for linear mixed model association of non-heritable phenotypes carried out with GCTA-LOCO for genotypes generated under the recent structure model.

cannot be detected using standard genomic inflation, which assumes that stratification impacts enough variants to shift the median of the test statistic (*Devlin and Roeder, 1999*), making it difficult to distinguish between true associations and residual stratification.

## Burden tests are relatively robust to local environmental effects

In practice, single rare variant association tests are often underpowered. To circumvent this, many studies aggregate information across multiple rare variants in a gene. Because they aggregate across rare variants, such tests have the potential to be affected by rare variant stratification (*Mathieson and McVean, 2012*). To study this, we examined the behavior of a simple gene burden statistic—the total number of rare derived alleles (frequency < 0.001) in each gene. We find that for a gene of average size (total exon length of ≈ 1.3 kb, mean of 16 rare variants), burden tests are robust to local effects under both perpetual and recent structure models (*Figure 3*). Because the burden statistic involves averaging over many variants, it behaves more like a common variant than a rare variant in terms of its spatial distribution (*Figure 3—figure supplement 1*). Thus, it is still susceptible to confounding by smoothly distributed environmental effects, but this can be corrected by common-PCA in the perpetual structure model or rare-PCA in either model (*Figure 3*).

More generally, the spatial distribution of gene burden depends on the number of variants and the recombination distance across which it is aggregated. Gene burden should become geographically less localized with an increase in the number of aggregated rare variants as each is likely to arise in an independent branch of the genealogy (*Figure 3—figure supplement 1*). As genetic distance between mutations increases, recombination decouples genealogies on which they arise, further reducing the probability of multiple mutations occurring on the same branch. Conversely, the

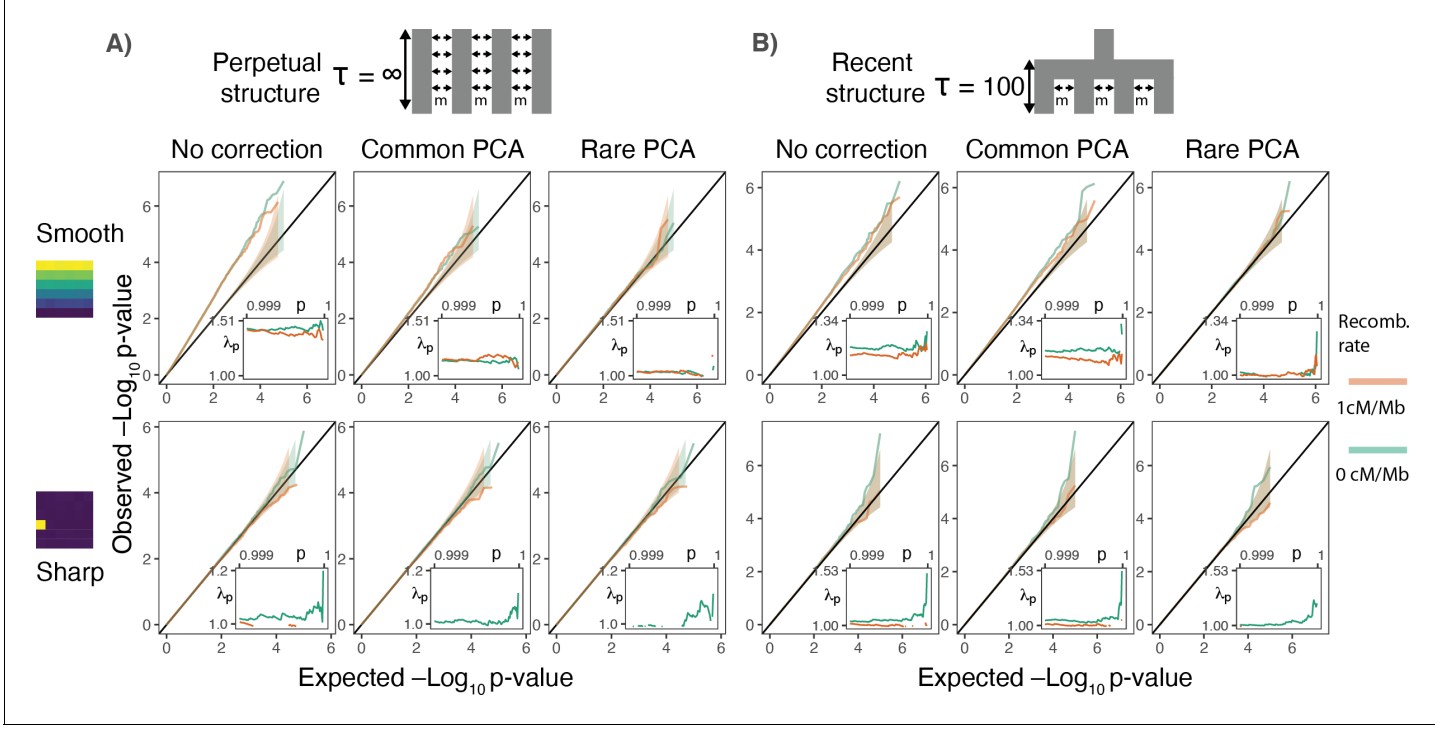

**Figure 3.** Gene burden tests are relatively robust to stratification. QQ plots of expected and observed -log₁₀p-value under the (A) perpetual and (B) recent structure models for the association of rare variant burden across a gene with total exon length of 1.3 kb (gene length of 7 kb) and non-heritable phenotype with a smooth (upper) or sharp (lower) distribution of environmental effects. Orange and green lines show results for a gene with and without recombination, respectively. Inset shows inflation in the tail (99.9%) of the test statistic distribution.

The online version of this article includes the following figure supplement(s) for figure 3:

**Figure supplement 1.** Gini curves showing geographic clustering of (A) variant frequency and (B) gene burden.

rare variant burden aggregated across few variants in genes with little recombination behaves more like a single rare variant and is susceptible to local effects (*Figure 3B* lower row).

## Polygenic scores capture residual environmental stratification

Polygenic scores—constructed by summing the effects of large numbers of associated variants—offer a simple way to make genetic risk predictions. At least in European ancestry populations, they can explain a substantial proportion of the phenotypic variance in complex traits like height (*Yengo et al., 2018*), BMI (*Yengo et al., 2018*), and coronary artery disease risk (*Khera et al., 2018*). However, their practical utility is limited by lack of transferability between populations (*Scutari et al., 2016*; *Martin et al., 2017*; *Kerminen et al., 2019*; *Wang et al., 2020b*) and between subgroups within populations (*Mostafavi et al., 2020*). This may be due in part to stratification in polygenic scores. To understand the behavior of polygenic scores under the perpetual and recent structure models, we simulated GWAS (N = 9000) of a heritable phenotype with a genetic architecture similar to that of height. We used GWAS effect sizes to calculate polygenic scores in an independent sample (N = 9000) and subtracted the true genetic values for each individual to examine the spatial bias in polygenic scores due to stratification.

Under both perpetual and recent structure models, residual polygenic scores are spatially structured, recapitulating environmental effects even when 100 common-PCs are used as covariates in the GWAS (*Figure 4*). LMMs perform similarly (*Figure 4—figure supplement 1*). This is due to the fact that when population stratification is not fully corrected, the effect sizes of variants that are correlated with the environment tend to be over- or under-estimated depending on the direction and strength of correlation (*Figure 4—figure supplement 2*). Stratification in residual polygenic scores is minimal when the causal variants are known, but not when the score is constructed from the most

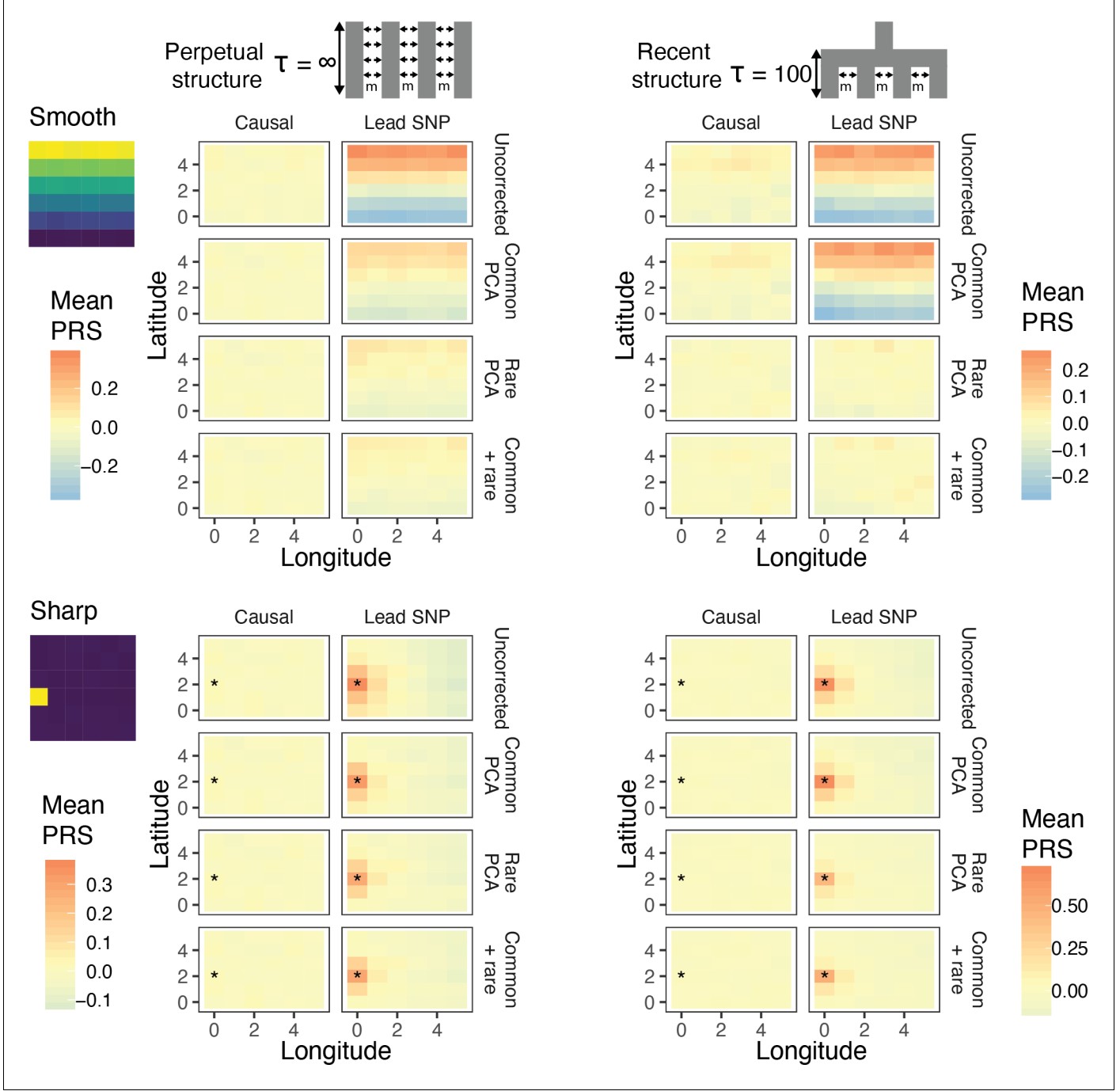

**Figure 4.** Residual stratification in effect size estimates translates to residual stratification in polygenic score in the (A) recent and (B) perpetual structure models. The simulated phenotype in the training sample has a heritability of 0.8, distributed over 2,000 causal variants. Each small square is colored with the mean residual polygenic score for that deme in the test sample, averaged over 20 independent simulations of the phenotype. In each panel, the rows represent different methods of PCA correction and columns represent two different methods of variant ascertainment. 'Causal' refers to causal variants with p-value $< 5 \times 10^{-4}$, and 'Lead SNP' refers to a set of variants, where each represents the most significantly associated SNP with a p-value $< 5 \times 10^{-4}$ in a 100 kb window around the causal variant. The simulated environment is shown on the left. For the sharp effect, the affected deme is highlighted with an asterisk.

The online version of this article includes the following figure supplement(s) for figure 4:

**Figure supplement 1.** Spatial distribution of residual polygenic scores based on effect sizes from linear mixed models carried out in GCTA-LOCO.

**Figure supplement 2.** Hexagonal bin plots illustrating the residual confounding in variant effect sizes due to stratification.

**Figure supplement 3.** Bias and prediction accuracy in polygenic scores as a function of variant ascertainment schemes.

significant SNPs ('lead SNPs') (*Figure 4*, *Figure 4—figure supplement 3*)—almost always the case in practice. Thus, picking the most significant SNPs (clumping and thresholding) tends to enrich for variants that are more structured than the causal variants, and improvements through statistical fine-mapping are marginal (*Figure 4—figure supplement 3*). Polygenic scores will be especially prone to residual stratification when constructed using SNPs that do not reach genome-wide significance. At such loci, the causal effects are likely to be small relative to the effect of stratification, leading to false identification of more structured variants.

## The effect of stratification in more complex models

In reality, genetic structure in most studies is more complex than either model discussed above. Most populations are genetically heterogeneous, and each genome is shaped by processes such as ancient and recent admixture, non-random mating, and selection, all of which vary both spatially and temporally. The present-day population of Britain, for example, is the result of a complex history of migration and admixture (*Leslie et al., 2015*; *Olalde et al., 2018*). Thus, restricting analysis even to the 'White British' subset of UK Biobank involves population structure on multiple time scales. To study these effects, we simulated under a model based on the demographic history of Europe and geographic structure of England and Wales, while maintaining the same degree of structure as the previous models (*Figure 5*, *Table 1*). In addition to recent geographic structure, we simulated an admixture event 100 generations ago between two populations, each of which are themselves the result of mixtures between several ancient populations (*Figure 5*). We varied the admixture fraction from the two source populations to create a North-South ancestry cline and sampled individuals to mimic uneven sampling in the UK Biobank (*Figure 5*, Materials and methods).

The results under this model are very similar to the recent structure model in that when the environmental effect is smoothly distributed, it cannot be corrected using common-PCA as population structure is largely recent (*Figure 5*). Note also that correction is not complete even with rare-PCA as seen from the biased polygenic scores of individuals from Cornwall, in the south-west of England (lower left deme in *Figure 5B*). This is not due to reduced migration in the region ('edge effects') but rather to uneven sampling (only 17 individuals sampled from Cornwall as opposed to 250 under uniform sampling). The bias disappears when individuals are sampled uniformly (*Figure 5—figure supplement 1*). Thus, our ability to correct for stratification and the utility of polygenic scores also depends on the sampling design of the GWAS. As with the other models, local effects cannot be corrected using either common- or rare-PCA (*Figure 5*).

## Polygenic scores based on effect sizes reestimated in siblings are not immune to stratification

Sibling-based studies test for association between siblings' phenotypic and genotypic differences. These, and other family-based association tests, are robust to population stratification as any difference in siblings' genotypes is due to Mendelian segregation and therefore uncorrelated with environmental effects. We simulated sibling pairs under the recent structure model and confirmed that polygenic scores constructed using SNPs and their effect sizes from the sibling-based tests were uncorrelated with environmental variation (*Figure 6* lower row).

In practice, however, sample sizes for sibling-based studies are much smaller than standard GWAS. A possible hybrid approach is to first ascertain significantly associated SNPs in a standard GWAS and then reestimate effect sizes in siblings. However, this approach is not completely immune to stratification. To demonstrate, we took the significant lead SNPs from a standard GWAS, reestimated their effect sizes in an independent set of 9,000 sibling pairs simulated under the same demographic model, and then generated polygenic scores in a third, independent, sample of 9,000 unrelated individuals. Polygenic scores generated this way are still correlated with the environmental effect when it is smoothly distributed, although less than when effect sizes from the discovery GWAS are used (*Figure 6*). Even though the sibling reestimated effects are unbiased, stratification in the polygenic score persists because the frequencies of the lead SNPs are systematically correlated with the environment. This is less pronounced for local effects because stratification is driven by variants that are rare in the discovery sample and often absent in the test sample (*Figure 6—figure supplement 1*).

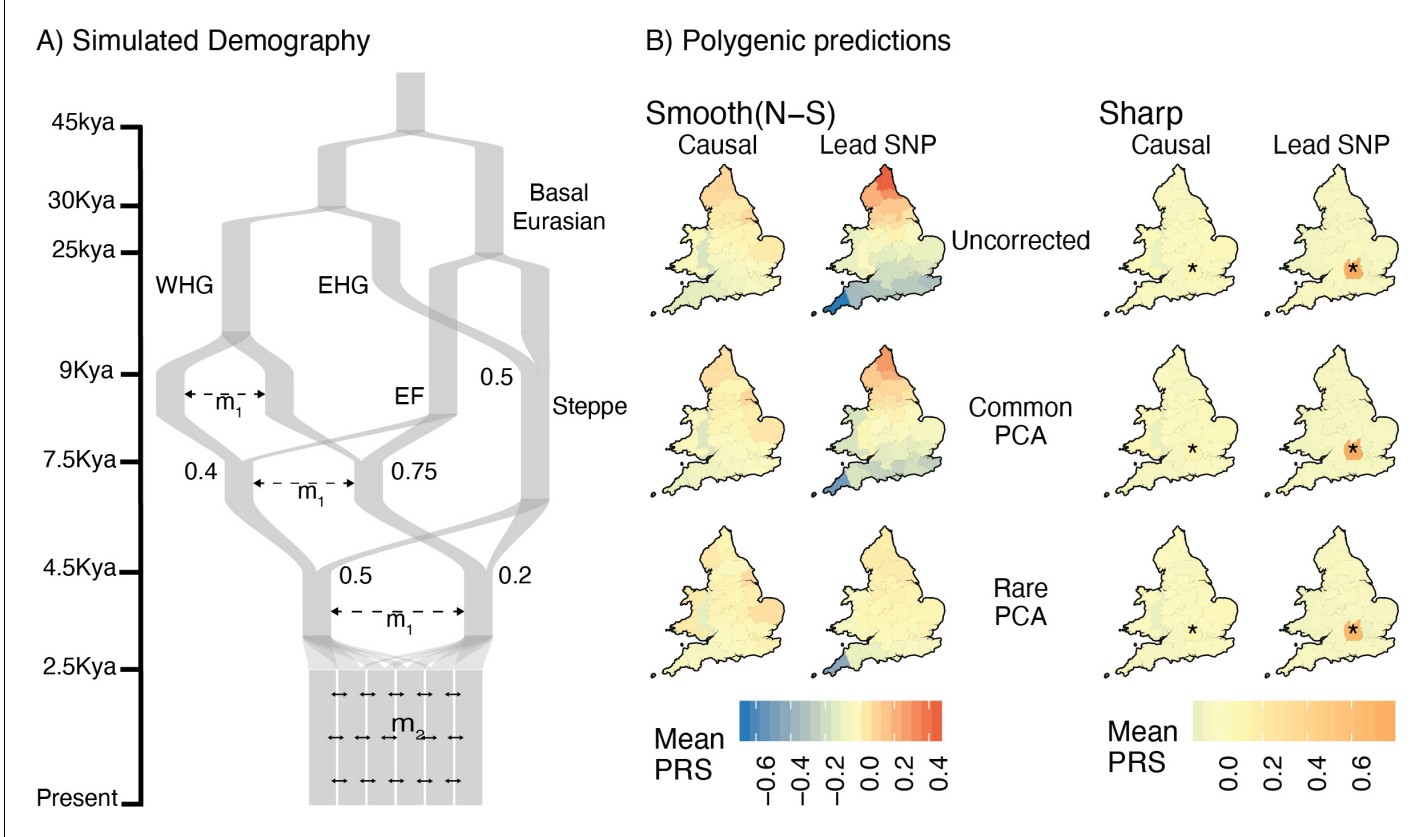

**Figure 5.** Residual stratification in polygenic scores under a complex demographic model, geographic structure representing England and Wales, and non-uniform sampling. (A) Illustration of the simulated demography. (B) Maps depicting the spatial distribution of residual polygenic scores, as in *Figure 4*, averaged across 20 simulations of the phenotype. Columns: 'Smooth' and 'Sharp' refer to environmental effects and 'Causal' and 'Lead SNP' refer to sets of variants that were used to construct polygenic scores. Rows: Different methods of correction for population structure. WHG and EHG: Western and Eastern Hunter Gatherers; EF: Early Farmers.

The online version of this article includes the following figure supplement(s) for figure 5:

**Figure supplement 1.** Spatial distribution of residual polygenic scores under the 'complex' structure model when individuals are sampled uniformly across demes.

**Figure supplement 2.** Example of how one might empirically minimize the inflation in test statistic in a genome-wide association study (GWAS) without knowledge of the demographic history.

**Figure supplement 3.** PCs computed from a single genetic relatedness matrix (GRM) constructed from common and rare variants together are different in how they capture population stratification than PCs computed from GRMs constructed separately from common and rare variants.

One argument in favor of the hybrid approach is that it balances the trade-off between bias and prediction accuracy. We show that the predictive accuracy of this approach is indeed higher than if both variants and effects were discovered in either standard or sibling GWAS (*Figure 6*). However, this is not an effect of the hybrid approach specifically but that of reestimation in general. Reestimating effect sizes in an independent cohort of unrelated individuals produces similar improvements in bias and prediction accuracy of polygenic scores (*Figure 6—figure supplement 2*).

## Discussion

The effect of population structure on GWAS depends on the amount of structure, the frequency of the variants tested and the distribution of confounding environmental effects. Here, we demonstrated that it also depends on the demographic history of the population in a way that is not fully captured by the degree of structure as summarized by $F_{ST}$ and genomic inflation. Consequently, to fully correct for population structure, it is necessary to know not only the degree of realized structure, but also the demographic history that generated it.

**Table 1.** Mean observed $F_{ST}$ for different migration rate under each demographic model.

| Model | Migration rate1 | Mean $F_{ST}$ (95% C.I.) | λ (Latitude) | λ (Longitude) |
| --- | --- | --- | --- | --- |
| Recent | 0.001 | 3.8e-03 (3.7e-03 - 4e-03) | 3.5649 | 3.7808 |
| Recent | 0.0025 | 2.6e-03 (2.5e-03–2.7e-03) | 3.4733 | 3.6425 |
| Recent | 0.005 | 1.6e-03 (1.5e-03–1.7e-03) | 3.0914 | 3.1357 |
| Recent | 0.0075 | 1.5e-03 (1.4e-03–1.6e-03) | 3.4661 | 3.3344 |
| Recent | 0.01 | 1.1e-03 (1e-03–1.2e-03) | 3.0629 | 3.0675 |
| Recent | 0.015 | 7.9e-04 (7.2e-04–8.6e-04) | 2.8256 | 2.5172 |
| Recent | 0.02 | 7e-04 (6.3e-04–7.7e-04) | 2.4668 | 2.6838 |
| Recent | 0.025 | 5.1e-04 (4.4e-04–5.9e-04) | 2.2173 | 2.6485 |
| Recent | 0.03 | 4e-04 (3.3e-04–4.6e-04) | 2.4842 | 2.2036 |
| Recent | 0.05* | 2.3e-04 (1.7e-04–2.9e-04) | 1.6754 | 1.8486 |
| Perpetual | 0.06 | 2.5e-04 (1.9e-04–3.1e-04) | 1.8101 | 1.7606 |
| Perpetual | 0.07* | 2.0e-04 (1.4e-04–2.6e-04) | 1.6640 | 1.6381 |
| Perpetual | 0.08 | 1.7e-04 (1.1e-04–2.3e-04) | 1.5905 | 1.6658 |
| Complex | 0.05 | 3.2e-04 (2.5e-04–3.8e-04) | 2.6425 | 1.7480 |
| Complex | 0.06 | 2.8e-04 (2.1e-04–3.4e-04) | 2.1651 | 1.8637 |
| Complex | 0.07 | 2.5e-04 (1.8e-04–3.1e-04) | 1.9318 | 1.7012 |
| Complex | 0.08* | 1.5e-04 (9.7e-05–2.1e-04) | 1.6520 | 1.5214 |
| Complex | 0.09 | 1.7e-04 (1.1e-04–2.2e-04) | 1.6841 | 1.3892 |
| Complex | 0.1 | 1.7e-04 (1.2e-04–2.3e-04) | 1.5943 | 1.4719 |
| Complex | 0.12 | 1.3e-04 (7.3e-05–1.8e-04) | 1.4442 | 1.4395 |
| Complex | 0.15 | 7.9e-05 (2.7e-05–1.3e-04) | 1.2536 | 1.3123 |

Proportion of migrants in and out of a deme per generation. Selected migration rate indicated with * for each model.

Generally, PCA (or mixed models) based on common variants will inadequately capture and correct population structure with a recent origin. This might partly explain why polygenic scores derived from studies such as the UK Biobank (*Haworth et al., 2019*; *Abdellaoui et al., 2019*) and FINNRISK (*Kerminen et al., 2019*) exhibit geographic clustering. In such cases, PCA based on rare variants, which are more informative about recent population history (*Gravel et al., 2011*; *Fu et al., 2013*; *O'Connor et al., 2013*; *O'Connor et al., 2015*; *Mathieson and McVean, 2015*), would be more effective. Haplotype sharing (*Lawson et al., 2012*) or identity-by-descent (IBD) segments are similarly informative about recent history (*Palamara et al., 2012*; *Ralph and Coop, 2013*; *Saada et al., 2020*), and provide an alternative to rare variant PCA when sequence data are not available, or when there are relatively few rare variants to adequately capture the structure, for example in exome sequence data.

This still leaves the question of exactly which frequency of variants (or length of IBD segments) to use. The structure in most studies exists on multiple time scales, even in relatively homogeneous populations (*Byrne et al., 2020*). In such cases, sets of PCs derived from variants in different frequency bins, or from IBD segments of different lengths, may be needed. PCs can be chosen based on visual inspection for significant axes of population structure (e.g. *Figure 5—figure supplement 2A–C*). However, even among the PCs that exhibit population structure, not all will contribute to the phenotype unless they are correlated with the confounding environmental effect(s), the distribution of which is *a priori* unknown. An empirical solution to this problem is to carry out a set of preliminary GWAS, each with different sets of PCs and use the summary statistics with the smallest inflation (*Figure 5—figure supplement 2D*). By letting the model learn the weights of PCs derived from different frequency bins, this approach has the added benefit of allowing for non-linearity in the contribution of stratification at different time scales. For example, under our complex model, using both

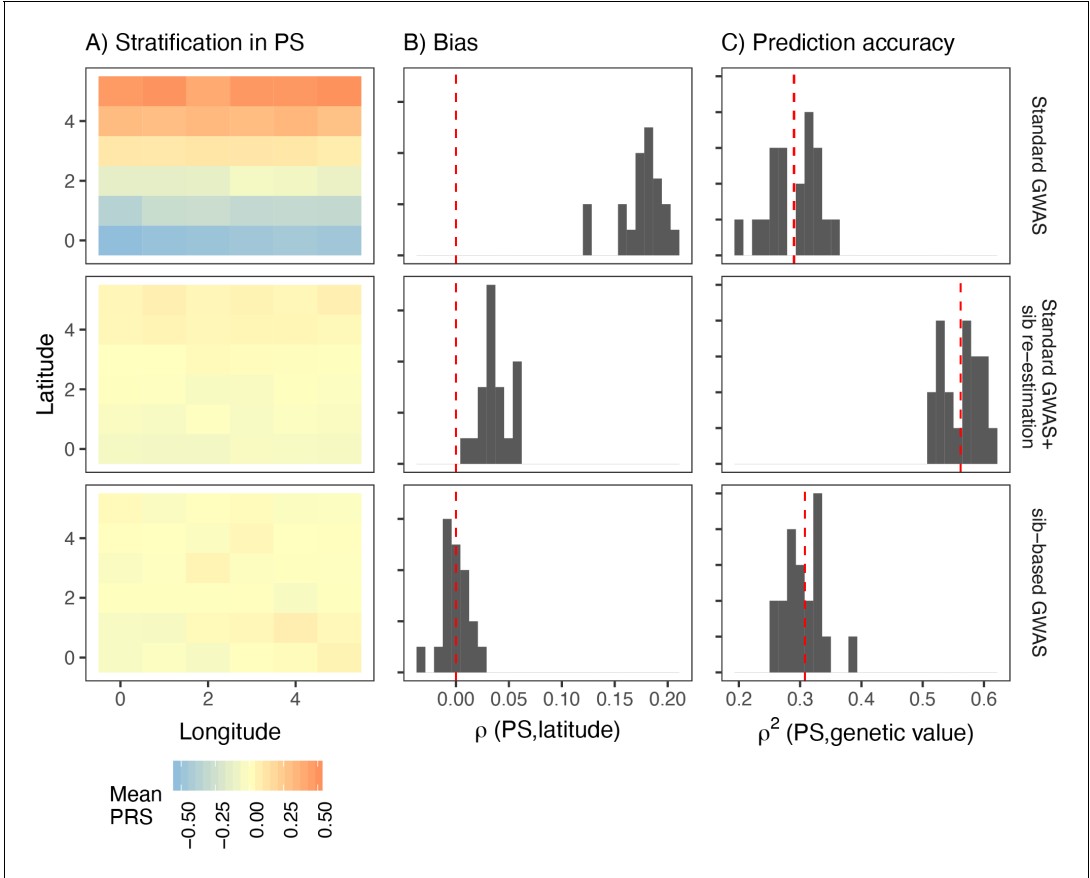

**Figure 6.** Comparison of stratification and predictive accuracy of polygenic scores between standard and sibling-based association tests under the recent structure model. Phenotypes simulated as in *Figure 4*. (A) Spatial distribution of polygenic scores generated using (top) effects of variants discovered in a standard genome-wide association study (GWAS; middle) variants ascertained in a standard GWAS but with effect sizes reestimated in sib-based design, (bottom) variants ascertained and effect sizes estimated in sib-based design. In each case, the effect is averaged over 20 simulations. (B) Bias and (C) predictive accuracy of polygenic scores for 20 simulations of the smooth environmental effect.

The online version of this article includes the following figure supplement(s) for figure 6:

**Figure supplement 1.** Spatial distribution of residual polygenic scores when the environment is sharply distributed (risk location indicated with *).
**Figure supplement 2.** Bias (A) and prediction accuracy (B) of polygenic scores calculated using different variant ascertainment and effect size estimation schemes.

common- and rare-PCs corrects for structure better than models where either rare- or common-PCs were used alone (*Figure 5—figure supplement 3*).

PCA- or LMM-based corrections are only effective when environmental effects are smoothly distributed with respect to ancestry or when they can be expressed as a linear function of the GRM. Sharply distributed effects (e.g. local environment or batch effects) may not be fully corrected with any method, regardless of the demographic history of the population. Such confounders are an important concern for rare variant studies. Because local effects lead to inflation in the tails of the test statistic distribution, single rare variant associations should always be treated with caution. Fortunately, burden tests are more robust to local effects than single rare variant tests, although, the degree to which burden statistics will be sensitive to local effects depends on the number of variants and the recombination distance between them—short genes with fewer variants will be more sensitive to local effects.

Even imperfect correction for population structure is probably sufficient to limit the number of genome-wide false positive associations in GWAS. But when information is aggregated across a large number of marginally associated variants, even small overestimates in effect sizes can lead to substantial bias in polygenic scores. Essentially some of the predictive power of polygenic scores will derive from predicting environmental structure rather than genetic effects. Comparison of polygenic

scores derived from standard GWAS and sibling-based studies suggests that this effect can be substantial (*Mostafavi et al., 2020*), and it may also contribute to inflated estimates of heritability and genetic correlation (*Browning and Browning, 2011*). Even though family-based studies are immune to stratification, we show that the practice of discovering associations in a standard GWAS and then reestimating their effects in siblings improves prediction and reduces, but does not eliminate, bias in polygenic scores if there is inadequate correction in the original GWAS. However, this is largely because of the advantages of reestimating effect sizes in a different sample, rather than specifically because of the use of siblings.

Our study focused on population structure arising from ancient admixtures and geographic structure because these are relatively well-understood and easy to model. However, our results generalize to any type of population structure, for example due to social stratification or assortative mating. What we refer to as local environmental effects also includes socially structured factors such as cultural practices. Ultimately, no single approach can completely correct for population stratification and replication in within-family studies and populations of different ancestry will provide greater confidence. To facilitate the evaluation of any residual population stratification in summary statistics, we recommend that studies report the following: (i) Summary statistics for all methods of correction attempted (e.g. PCA or LMMs where the GRM is constructed from variants in different frequency bins); (ii) Summary statistics for association with any available demographic variables such as birthplace (e.g. *Haworth et al., 2019*); (iii) Summaries of the distribution of polygenic scores (for a subset of the data not used in the original GWAS) with respect to geography, ancestry, and principal components (e.g. *Kerminen et al., 2019*). These summaries will be helpful for downstream evaluation of the robustness of polygenic predictions.

## Materials and methods

### Simulations of population structure

We used *msprime* (*Kelleher et al., 2016*) to simulate genotypes in a 6×6 grid of demes and modeled the demographic history in three different ways: (i) where the structure extends infinitely far back in time ('perpetual'), (ii) where all demes collapse into a single population 100 generations in the past ('recent'), and (iii) a more complex model that is loosely based on the demographic history of Europe (*Lazaridis, 2018*; *Figure 5*; 'complex'). We fixed the effective population size of all demes and the merged ancestral population sizes to 10,000 diploid individuals.

For the perpetual and recent models, we parameterized the degree of structure in the data with a fixed, symmetric migration rate among demes ($m$) chosen to match the degree of structure observed in Britain. To select an appropriate value for $m$, we simulated a 10 Mb genome (10 chromosomes of 1 Mb each) with mutation and recombination rates of $1 \times 10^{-8}$ per-base per-generation, for 9,000 individuals (250 per-deme) for a range of migration rates under each demographic model (*Table 1*). We estimated mean $F_{ST}$ across all demes with the Weir and Cockerham estimator (*Weir and Cockerham, 1984*) using an LD-pruned (PLINK –indep-pairwise 100 10 0.1; *Purcell et al., 2007*; *Chang et al., 2015*) set of common variants (MAF > 0.05). We used the ratio of averages approach (*Bhatia et al., 2013*) to calculate $F_{ST}$ and estimated genomic inflation on birthplace ($\lambda_{location}$) by carrying out GWAS on an individual's $x$ and $y$ coordinates in the grid, similar to the GWAS on longitude and latitude in *Haworth et al., 2019*. The migration rate was chosen for each model separately to roughly match the mean $F_{ST}$ observed among regions in Britain ($\approx 0.0007$) (*Leslie et al., 2015*) and $\lambda_{location} \approx 12$ reported for the UKB (*Haworth et al., 2019*). Because genomic inflation scales linearly with sample size (*Bulik-Sullivan et al., 2015b*), we matched the expected value given our sample size of 9K using:

$$\lambda_{location}^{9k} = \frac{9}{300}(\lambda_{location}^{300k} - 1) + 1 \tag{1}$$

Where $\lambda_{location}^{300k}$ is the observed value ($\approx 12$) given a sample size of 300,000 as in *Haworth et al., 2019*. Plugging this in, we get an expected value of $\lambda_{location}^{9k} \approx 1.36$. To match this approximately, we set the migration rate to a fixed value of 0.05 and 0.07 for the 'recent' and 'perpetual' models, respectively (*Table 1*).

We parameterized the 'complex' model with two migration rates, $m_1$ and $m_2$, where $m_1$ represents the migration rate between the source populations mixing 100 generations before present (2.5kya) and $m_2$ represents the migration rate between adjacent demes in the grid (*Figure 5*). We selected $m_1$ and $m_2$ in a step-wise manner, first setting $m_1 = 0.004$ (representing the $F_{ST}$ between the two source populations) to match the maximum $F_{ST}$ between regions in Britain. We then set $m_2 = 0.08$ (representing subsequent mixing and isolation by distance) to match the mean $F_{ST}$ between regions in Britain (*Leslie et al., 2015*; *Table 1*). In all cases, after selecting the appropriate migration parameters, we re-simulated genotypes under each model for a larger genome of 200 Mb (20 chromosomes of 10 Mb each), which we used for all further analysis.

## Geographic structure in England and Wales

We downloaded the Nomenclature of Territorial Units for Statistics level 2 (NUTS2) map for 35 regions in England and Wales (version 2015) from data.gov.uk and assigned each individual of 'White British' ancestry in the UKB to a region based on their birthplace. We calculated the proportion of individuals sampled from each region and used these as weights in our simulations to mimic the sampling distribution in the UKB. To generate a migration matrix between regions, we generated an adjacency matrix for the NUTS2 districts using the 'simple features' (sf) R package (*Pebesma, 2018*), where an entry is one if two districts abut and zero otherwise, and multiplied this matrix by the migration parameter $m_2$.

## Simulation of phenotypes

To study the effect of stratification on test statistic inflation, we simulated non-heritable phenotypes $y_{ij}$ of an individual $i$ from deme $j$ as $y_{ij} \sim N(\mu_j, \sigma)$, where $\mu_j$ is the mean environmental effect in deme $j$. For the smooth effect, we chose $\mu_j$ such that the difference between the northern and southern-most demes was $2\sigma$. For the sharp effect, we set $\mu_j = 2\sigma$ for one affected deme and zero otherwise. To test the impact of population structure on effect size estimation and polygenic score prediction, we simulated heritable phenotypes using the model described in *Schoech et al., 2019*. We selected 2,000 variants across the 200 Mb genome (one variant chosen uniformly at random in each 100 kb window) and sampled their effect sizes as $\beta_k \sim N(0, \sigma_l^2[p_k(1-p_k)]^\alpha)$ where $\sigma_l^2$ is the frequency-independent component of genetic variance, $p_k$ is the allele frequency of the $k^{th}$ variant, and $\alpha$ is a scaling factor. We set $\alpha = -0.4$ based on an estimate for height (*Schoech et al., 2019*) and $\sigma_l^2$ such that the overall genetic variance underlying the trait, $\sigma_g^2 = \sigma_l^2 \sum_{k=1}^{M}[2p_k(1-p_k)]^{\alpha+1} = 0.8$. We calculated the genetic value for each individual, $g_i = \sum_{k=1}^{M} \beta_k x_{ik}$, where $x_{ik}$ is the number of derived alleles individual $i$ carries at variant $k$, and added environmental effects as described above. We generated 20 random iterations of both heritable and non-heritable phenotypes.

## GWAS

We simulated 18,000 individuals (500 from each deme) under each demographic model and split the sample into two equally sized sets, a training set on which GWAS and PCA were carried out, and a test set for polygenic score predictions. Common-PCA and rare-PCA were carried out using PLINK (*Chang et al., 2015*) on a set of 200,000 common (MAF > 5%) and one million rare (minor allele count = 2, 3, or 4) variants, respectively, sampled from all variants generated under each model. To carry out PCA on identity-by-descent (IBD) sharing, we called long (>10 cM) pairwise IBD segments using GERMLINE (*Gusev et al., 2009*) with default parameters and generated an IBD-sharing GRM, in which each entry represents the total fraction of the haploid genome (100 Mb) shared by individual pairs. We calculated eigenvectors (PCs) of the IBD-sharing GRM using GCTA (*Yang et al., 2011*).

We performed GWAS using –glm in PLINK 2.0 with 100 PCs as covariates (*Chang et al., 2015*). As indicated in the main text, we also used as a set of 50 common- and 50 rare-PCs, computed separately, as covariates in the same model to correct for structure existing on multiple time scales.

We fitted LMMs using GCTA-LOCO (*Yang et al., 2011*) where the GRM was based on the same common or rare variants used for PCA. GCTA's LOCO (leave one chromosome out) algorithm fits a model where the GRM is constructed from SNPs that are not present on the same chromosome as the variant being tested to avoid proximal contamination. We also included the top 100 PCs as fixed effects in the mixed models.

We calculated genomic inflation ($\lambda_p$) for non-heritable phenotypes as $\frac{\chi_p^2}{F_{\chi^2}^{-1}(p)}$ where $\chi_p^2$ is the $p^{th}$ percentile of the observed association test statistic and $F_{\chi^2}^{-1}(p)$ is the quantile function of the $\chi^2$ distribution with 1 degree of freedom.

### Sibling-based tests

We conducted structured matings by sampling pairs of individuals from the same deme and generated the haplotypes of each child by sampling haplotypes, with replacement, from each parent without recombination. We generated heritable phenotypes as described in the previous section for each sibling and modeled the effect of each variant as

$$\Delta y_i = \beta_i \Delta x_i + \epsilon_i$$

where $\Delta y$ is the difference in siblings' phenotypic values and $\Delta x_i$ is the difference in the number of derived alleles at the $i^{th}$ variant.

### Polygenic scores

We calculated polygenic scores for each individual as $\sum_i \hat{\beta}_i x_i$ where $\hat{\beta}_i$ is the estimated effect size and $x_i$ is the number of derived alleles for the $i^{th}$ variant (either causal or lead SNP). To study patterns of *residual* stratification, we subtracted individuals' true (simulated) genetic values ($g_i = \sum_i \beta_i x_i$), which themselves can be structured, from polygenic scores. We averaged residual polygenic scores across 20 random iterations of causal variant selection, effect size generation, and GWAS to minimize stochastic variation. Predictive accuracy of polygenic scores was measured as the proportion of variance in individuals' genetic values that can be explained by their polygenic score.

### Gene burden

We simulated genes, each with eight exons of length 160 bp separated by introns of length 6,938 bp, representing an average gene in the human genome (*Piovesan et al., 2019*). We simulated 100,000 genes for the 'recent' model with and without recombination and for the 'perpetual' model with no recombination. For the 'perpetual' model with recombination, we simulated 50,000 genes. We calculated gene burden as the total count of derived alleles (frequency < 0.001) across all exons in the gene for each individual. Even though introns do not directly contribute to gene burden, they serve as spacers to allow for recombination between exons. In genes without recombination, introns only add to the computational cost and, therefore, we did not simulate them. To ensure that differences in structure in gene burden between models was driven by differences in demographic history and not differences in the number of rare variants, we first calculated the mean (16) and standard deviation (4) of the number of rare variants under the 'recent' model and sampled from this distribution when simulating under the 'perpetual' model. The geographic clustering of burden was measured using Gini curves and the Gini coefficient.

$$G = \frac{n - y_1 - \sum_{1 < i \leq n} (y_i + y_{i-1})}{n}$$

where $y_i$ is the cumulative gene burden in the $i^{th}$ deme sorted in increasing order of gene-burden and $n$ is the number of demes. The Gini coefficient ranges from zero, indicating that the burden is uniformly distributed in space, to one, indicating that the burden is concentrated in a single deme (*Figure 3—figure supplement 1*).

### Imputation and fine-mapping

We performed imputation using Beagle 5.1 (*Browning et al., 2018*). We imputed the genotypes of rare variants (MAF < 0.001) in a sample of 9,000 individuals using the phased sequences of an independent 9,000 individuals as reference. Both reference and test sets were simulated under the recent structure model.

We fine-mapped variants using SuSiE (*Wang et al., 2020a*) separately on 100 Kb windows, each of which carried a single causal variant. We restricted fine-mapping to windows where at least one

variant had a p-value $< 1 \times 10^{-4}$ and picked the variant with the highest posterior inclusion probability to construct polygenic scores.

## Code availability

We carried out all analyses with code written in Python 3.5, R 3.5.1, and shell scripts, which are all available at https://github.com/Arslan-Zaidi/popstructure; *Zaidi, 2020*; copy archived at swh:1:rev: 1509a53ee491e3e01320c174ff55f9426da8923f.

## Acknowledgements

This research was supported by NIGMS award number R35GM133708. The content is solely the responsibility of the authors and does not necessarily represent the official views of the National Institutes of Health. The UK Biobank Resource was used under Application 33923.

## Additional information

### Funding

| Funder | Grant reference number | Author |
| --- | --- | --- |
| National Institute of General Medical Sciences | R35GM133708 | Iain Mathieson |

The funders had no role in study design, data collection and interpretation, or the decision to submit the work for publication.

### Author contributions

Arslan A Zaidi, Conceptualization, Formal analysis, Investigation, Visualization, Methodology, Writing - original draft, Writing - review and editing; Iain Mathieson, Conceptualization, Supervision, Funding acquisition, Investigation, Methodology, Project administration, Writing - review and editing

### Author ORCIDs

Arslan A Zaidi (iD) https://orcid.org/0000-0002-2155-8367

### Decision letter and Author response

Decision letter https://doi.org/10.7554/eLife.61548.sa1
Author response https://doi.org/10.7554/eLife.61548.sa2

## Additional files

### Supplementary files

• Transparent reporting form

### Data availability

The data used in this study were generated through simulations. The code for these simulations is freely available at https://github.com/Arslan-Zaidi/popstructure (copy archived at https://archive.softwareheritage.org/swh:1:rev:1509a53ee491e3e01320c174ff55f9426da8923f/) and can be used to reproduce all simulations and carry out all analyses in the manuscript.

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
