## [Decision Letter]

**Acceptance summary:**

A major current challenge in human genetics is the effect that population structure can have on results from genome-wide association studies and the applications thereof, including with polygenic scores for trait variation or disease risks. Uncorrected biases may be subtle at the level of individual genotype-phenotype associations but can still have meaningfully large effects on an additive basis for a complex trait. The present study from Zaidi and Mathieson meaningfully advances the field by both demonstrating that recent population structure cannot be corrected effectively via a common approach and presenting potential solutions.

**Decision letter after peer review:**

Thank you for submitting your article "Demographic history impacts stratification in polygenic scores" for consideration by *eLife*. Your article has been reviewed by three peer reviewers, and the evaluation has been overseen by George Perry as the Senior and Reviewing Editor. The following individual involved in review of your submission has agreed to reveal their identity: Alicia R Martin (Reviewer #3).

The reviewers have discussed the reviews with one another and the Reviewing Editor has drafted this decision to help you prepare a revised submission.

As the editors have judged that your manuscript is of interest, but as described below that additional analyses are required before it is published, we would like to draw your attention to changes in our revision policy that we have made in response to COVID-19 (https://elifesciences.org/articles/57162). First, because many researchers have temporarily lost access to the labs, we will give authors as much time as they need to submit revised manuscripts. We are also offering, if you choose, to post the manuscript to bioRxiv (if it is not already there) along with this decision letter and a formal designation that the manuscript is "in revision at *eLife*". Please let us know if you would like to pursue this option. (If your work is more suitable for medRxiv, you will need to post the preprint yourself, as the mechanisms for us to do so are still in development.)

Summary:

This paper provides a strong and clear advance on the issue of population stratification in human genome-wide association studies and the downstream biases that arise from it. The authors reaffirm that while such biases can be subtle regarding individual SNP effects, polygenic scores nonetheless accumulate large errors genome-wide. Through simulations of various realistic demographic models, they show that no single approach they tested completely corrects for population structure, indicating that complex demographic considerations are required to elucidate the role of stratification on polygenic scores. Principal components calculated from rare variants appear to better capture recent population structure than common variants, such that a multivariate approach with multiple sets of PCs may be superior.

Essential revisions:

1) Demographic parameters: There is immense interest in clinical applications of PRS, but as this and previous work has shown, FST between discovery and target cohorts is only the first indicator of issues in translation. In the Discussion, the authors should consider proposing more comprehensive frameworks for assessing stratification beyond FST. For example, is there a standard approach that the field could use to quantify stratification in GWAS summary statistics (e.g. in comparison to a reference panel)? Since FST itself is insufficient, what demographic parameters or metrics in the discovery and target cohorts could be reported to facilitate translation?

2) Effects of imputation: The results reported here indicate that GWAS summary statistics contain less population stratification when PCs are calculated using rare variants. This solution seems great in theory. In practice, concerns with this approach in GWAS studies may arise because of varying imputation accuracy as a function of ancestry and allele frequency, with especially low imputation accuracy among rarer variants and in underrepresented ancestries. It would be very helpful to use the simulation framework and data here to determine the extent to imputation errors impact PRS accuracy in the UK Biobank when including PCs from more accurately imputed common versus less accurately imputed rare variants.

(3a) One proposal in the Discussion is that considering two sets of PCs in a standard linear regression or linear mixed model may reduce residual stratification. How collinear are PCs computed from common and rare variants, e.g. in Figures 1, 2, 4, 6, Figure 1—figure supplement 2, Figure 2—figure supplement 1, Figure 4—figure supplements 1 and 2, Figure 5—figure supplement 1 and Figure 3—figure supplement 1, particularly as a function of number of SNPs used in their calculation? Further analysis would be helpful to guide whether and the extent to which there is a tradeoff between stratification and power differences from collinearity and # degrees of freedom. (This may have consequences for study design, e.g. GWAS arrays vs. exome or genome sequencing).

(3b) Is it actually important that the PCs be obtained by independent eigendecomposition/SVD on variants from different frequency bins? Alternatively, would it be sufficient to just make sure to include variants of different frequency classes in the genotype matrix, and then get a single set of PCs from the combined set? E.g. if you combine the common and rare variants into a single genotype matrix and then include the top 200 PCs from that matrix, does this approach perform equally as well as the one where you independently get the top 100 PCs each from common and rare? Some care would need to be taken to make sure this comparison was done fairly, as you'd want to make sure that the common and the rare variants explained an equal amount of variance in the top 200 PCs, mimicking the situation where you've provided an equal number of rare and common PCs. Given that PCA is a linear procedure, the answer to this question seems like it would depend on whether the decision to split the genotype matrix by frequency bin before doing the PCA(s) represents some important non-linearity in your model of population structure. If this is indeed the case, it seems like breaking out of the linear constraint of PCA would be a more general path forward, and that would seem worth noting. If the combined approach can indeed match the approach of performing PCA separately, then it suggests that it's just a matter of making sure certain patterns are represented in some way in the underlying genotype matrix, and that, also, would seem worth noting.

4) Previous work, e.g. by Kerminen et al., 2019, has shown reduced overprediction across geographical regions when using mixed models. As this manuscript further considers PCs and LMMs as a function of allele frequency, more guidance regarding which PCs and GRM(s) should be included based on rare and/or common variants to minimize stratification would be helpful.

5) Fine-mapping: How much does fine-mapping have the potential to help? E.g. if we use state-of-the-art fine-mapping methods like SuSiE that produce posterior probabilities, can we diminish PRS stratification from lead SNP effects, and to what extent (maybe dependent on demographic history and sample size)?

6) Siblings: We agree with the authors' statement that ascertaining SNPs in the usual way and re-estimating effect estimates in siblings is not immune to stratification (Figure 5, subsection “Sibling-based tests are robust to environmental stratification”). In addition to stratification, there is also most likely also a tradeoff in accuracy. With these different strategies and tradeoffs in mind, in addition to correlation between polygenic scores and latitude, it would also be helpful to know how correlation between polygenic scores and phenotype vary with different SNP selection and effect size estimation strategies (e.g. in an additional panel C).

7) To help round out the manuscript, we would like the authors to add one or more examples based on their simulation results to illustrate how strategies they propose for dealing with uncorrected, residual population structure would actually work.

---

## [Author Response]

Essential revisions:1) Demographic parameters: There is immense interest in clinical applications of PRS, but as this and previous work has shown, FST between discovery and target cohorts is only the first indicator of issues in translation. In the Discussion, the authors should consider proposing more comprehensive frameworks for assessing stratification beyond FST. For example, is there a standard approach that the field could use to quantify stratification in GWAS summary statistics (e.g. in comparison to a reference panel)? Since FST itself is insufficient, what demographic parameters or metrics in the discovery and target cohorts could be reported to facilitate translation?

It’s certainly true that a single number like F_ST_ or even a multivariate measure like PCA can’t capture__ the complexity of population structure particularly since, as we show, this structure changes over time (and consequently depends on allele frequency). Therefore, we suggest that rather than trying to summarize population structure and demography, studies instead try to summarize the empirical effects of population structure by investigating the behaviour of different association tests. Some suggestions (which we now discuss in the Discussion) are:

a) Report summary statistics for association with demographic variables such as birthplace and ethnicity. A nice example of this was presented in Haworth et al., 2019, where they reported the genomic inflation in GWAS on birthplace in the UK Biobank.

b) Report summary statistics using PCA/LMMs where the GRM is constructed from variants of different frequencies (see more detailed response to point 4 below).

c) Report genomic inflation (for multiple quantiles) stratified by frequency of the variant.

d) Genomic inflation in summary statistics may be too subtle to be observed with standard methods (e.g. LD score regression) if stratification impacts the test statistic distribution non-uniformly (e.g. when environmental effects are sharply distributed). In such cases, the distribution of polygenic scores with respect to genetic principal components or geography (e.g. Haworth et al., 2019, Sohail et al., 2019, Berg et al., 2019, Kerminen et al., 2019, and Abdellaoui et al., 2018) could be an indicator of residual stratification in the summary statistics, though difficult to distinguish from true population differences in genetic value.

Ultimately, it is hard to completely rule out stratification. But, by reporting a more detailed set of association metrics, it will be possible to get a better sense of how much stratification is plausible in any particular analysis.

2) Effects of imputation: The results reported here indicate that GWAS summary statistics contain less population stratification when PCs are calculated using rare variants. This solution seems great in theory. In practice, concerns with this approach in GWAS studies may arise because of varying imputation accuracy as a function of ancestry and allele frequency, with especially low imputation accuracy among rarer variants and in underrepresented ancestries. It would be very helpful to use the simulation framework and data here to determine the extent to imputation errors impact PRS accuracy in the UK Biobank when including PCs from more accurately imputed common versus less accurately imputed rare variants.

To investigate how imputation accuracy of rare variants impacts rare-PCA, we used Beagle to impute rare variants (MAF < 0.001) using phased genotypes of common variants (MAF > 0.1) in a set of 9,000 individuals. We used the phased sequences of an independent set of 9,000 individuals as reference. Even though the imputation accuracy of rare variants is indeed lower than that of common variants (e.g. see Figure 1—figure supplement 2A), imputed rare variants capture population structure just as effectively as the un-imputed variants (Figure 1—figure supplement 2B). However, our reference panel was simulated under the same demographic history as the imputation set, which may not be realistic, especially for populations where appropriate reference panels are not available. In general, whether or not you can use imputed rare variants depends on how well you can impute them–an empirical question which is very dataset-dependent. We also suggest alternatives (e.g. PCA on haplotype- or IBD-sharing, Figure 1—figure supplement 2C-D) that may be more robust when appropriate reference panels or large sequencing data are not available. We now summarize this point in the Discussion and in (Figure 1—figure supplement 2).

(3a) One proposal in the Discussion is that considering two sets of PCs in a standard linear regression or linear mixed model may reduce residual stratification. How collinear are PCs computed from common and rare variants, e.g. in Figures 1, 2, 4, 6, Figure 1—figure supplement 2, Figure 2—figure supplement 1, Figure 4—figure supplements 1 and 2, Figure 5—figure supplement 1 and Figure 3—figure supplement 1, particularly as a function of number of SNPs used in their calculation? Further analysis would be helpful to guide whether and the extent to which there is a tradeoff between stratification and power differences from collinearity and # degrees of freedom. (This may have consequences for study design, e.g. GWAS arrays vs. exome or genome sequencing).

The collinearity between common and rare-PCs will depend on demographic history. We now discuss this in the main text in the Results section. In Figure 1—figure supplement 1, we show the proportion of variance in each rare-PC explained by the first 50 common-PCs (and vice versa) under the three demographic models. Under the perpetual structure model, the r^2^ ^^ between common and rare-PCs is quite high ( ~ 0.5) because common and rare variants capture the same structure, which is constant in time. On the other hand, in the recent structure models, common-PCs explain only a small fraction ( ~ 0.03) of the variance in the first rare-PC. The complex structure model is intermediate. This is essentially the same intuition that we get from looking at the principal component plots: In the recent structure model rare and common-PCs are distinct, in the perpetual structure model they are similar, and reality is intermediate. Finally we feel that the number of PCs is generally not large enough (compared to the sample size) for the loss of degrees of freedom to be a concern.

(3b) Is it actually important that the PCs be obtained by independent eigendecomposition/SVD on variants from different frequency bins? Alternatively, would it be sufficient to just make sure to include variants of different frequency classes in the genotype matrix, and then get a single set of PCs from the combined set? E.g. if you combine the common and rare variants into a single genotype matrix and then include the top 200 PCs from that matrix, does this approach perform equally as well as the one where you independently get the top 100 PCs each from common and rare? Some care would need to be taken to make sure this comparison was done fairly, as you'd want to make sure that the common and the rare variants explained an equal amount of variance in the top 200 PCs, mimicking the situation where you've provided an equal number of rare and common PCs. Given that PCA is a linear procedure, the answer to this question seems like it would depend on whether the decision to split the genotype matrix by frequency bin before doing the PCA(s) represents some important non-linearity in your model of population structure. If this is indeed the case, it seems like breaking out of the linear constraint of PCA would be a more general path forward, and that would seem worth noting. If the combined approach can indeed match the approach of performing PCA separately, then it suggests that it's just a matter of making sure certain patterns are represented in some way in the underlying genotype matrix, and that, also, would seem worth noting.

We now discuss this point in –the third paragraph of the Discussion. Typically, the genotypes are scaled by the inverse variance (12f(1−f)) of the SNP before the GRM is constructed. This tends to up-weight the contribution of rare variants, which are far greater in number. Therefore, PCs calculated using both common and rare variants together are similar to rare-PCs in how they capture population structure (Figure 5—figure supplement 3). Using sets of PCs calculated separately from common and rare variants allows some non-linearity in the relationship between stratification and frequency, allowing the model to learn the appropriate weighting of rare vs. common variants. In the complex structure model, we find that using 50 common + 50 rare performs better than 100 (common+rare) PCs, but this also depends on the environmental effect. That the environmental effect is *a priori* unknown is another reason to allow more flexibility in the model fit.

4) Previous work, e.g. by Kerminen et al., 2019, has shown reduced overprediction across geographical regions when using mixed models. As this manuscript further considers PCs and LMMs as a function of allele frequency, more guidance regarding which PCs and GRM(s) should be included based on rare and/or common variants to minimize stratification would be helpful.

It is difficult to generalize which set of variants should be used because this depends on (i) the demographic history of the sample, which can be inferred from genetic and historical data, but also on (ii) the distribution of confounding environmental effects, which we do not know *a priori*. One approach that we like is to carry out multiple GWAS, each with PCs (or a combination of PCs) computed from variants in different frequency bins and use the summary statistics with the smallest inflation (Figure Author response image 1) – thus empirically optimizing the efficiency of the correction. This approach could also be extended to multiple sets of PCs, for example as described in our response to point 2. We discuss this approach in the Discussion, and (Figure 5—figure supplement 2D).

**Author response image 1. respfig1:** Genomic inflation (y-axis) as a function of variant frequency used to calculate PCs (x-axis). The GWAS were carried out using genotypes generated under the recent structure model and the smooth phenotype.

In our simulations, when the population structure could be represented in the GRM, both PCs and LMMs performed similarly. This is also consistent with Kerminen et al., 2019, where the reduction in overprediction was only marginally better with LMMs. The original intent behind LMMs was to correct for background genetic effects (e.g. in controlled laboratory crosses) with the implicit assumption of an infinitesimal genetic architecture. Recent methods such as Bolt-LMM are flexible to the infinitesimal assumption but still model the confounding environment as a simple linear function of the GRM. PCA-based approaches are more flexible in that they allow environmental effects (of which there may be many) to be modeled by multiple parameters (the same as the number of PCs used in the model). Thus, our view is that PCA-based corrections are preferable to LMMs when the distribution of environmental effects are unknown — almost always the case. However, further work is required to study the efficacy of each approach under different demographic histories and environmental effects.

5) Fine-mapping: How much does fine-mapping have the potential to help? E.g. if we use state-of-the-art fine-mapping methods like SuSiE that produce posterior probabilities, can we diminish PRS stratification from lead SNP effects, and to what extent (maybe dependent on demographic history and sample size)?

If the causal variants were known precisely then, as we show, there is little effect of stratification. So this question is really an empirical one about how well fine mapping approaches can identify causal variants. As suggested, we ran SuSiE on our simulated GWAS and used the effect sizes of variants with the highest posterior probability to calculate polygenic scores. However, this approach improves prediction accuracy and reduces bias only very slightly compared to the approach where we used the most significant hits. We describe this analysis in the subsection “Polygenic scores capture residual environmental stratification” and in Figure 4—figure supplement 3. Why this is the case is a question on the efficacy of statistical fine-mapping, which we think needs further investigation.

6) Siblings: We agree with the authors' statement that ascertaining SNPs in the usual way and re-estimating effect estimates in siblings is not immune to stratification (Figure 5, subsection “Sibling-based tests are robust to environmental stratification”). In addition to stratification, there is also most likely also a tradeoff in accuracy. With these different strategies and tradeoffs in mind, in addition to correlation between polygenic scores and latitude, it would also be helpful to know how correlation between polygenic scores and phenotype vary with different SNP selection and effect size estimation strategies (e.g. in an additional panel C).

The prediction accuracy is indeed greater for the hybrid approach (standard discovery + sibling re-estimation) relative to either the standard GWAS or sib-based GWAS (Figure 6). Further investigation shows that this is not an effect of re-estimation in siblings but because re-estimation, in general, produces more unbiased effect size estimates. In fact, the increase in prediction accuracy with re-estimation is even greater than the increase in accuracy with a larger sample size (Figure 6—figure supplement 2). This suggests that perhaps one easy way to reduce the bias-accuracy tradeoff is to split GWAS into discovery and re-estimation sets. We thank the reviewer for suggesting this analysis and we have incorporated the results in Figure 6 and Figure 6—figure supplement 2, and into the subsection “Polygenic scores based on effect sizes re-estimated in siblings are not immune to stratification”.

7) To help round out the manuscript, we would like the authors to add one or more examples based on their simulation results to illustrate how strategies they propose for dealing with uncorrected, residual population structure would actually work.

We have now made suggestions in –the Discussion alongside edits suggested in point 4. Briefly, we recommend that researchers first explore the population structure in their data, which for most studies will exist on multiple timescales, by carrying out PCA on variants in different frequency bins or IBD segments of varying lengths. Then, carry out a set of preliminary GWAS, each with different sets of PCs and use the summary statistics with the smallest inflation. To reduce computational burden, these GWAS can first be carried out on a subset of variants uniformly sampled from the genome to find the optimal combination of PCs (e.g. based on genomic inflation) before carrying out a full GWAS on all variants (see Figure 5—figure supplement 2).